# Assessing ventilation through ambient carbon dioxide concentrations across multiple healthcare levels in Ghana

Cecilia Crews[1], Paul Angwaawie[2], Alhassan Abdul-Mumin[3,4], Iddrisu Baba Yabasin[5], Evans Attivor[6], John Dibato[7], Megan P. Coffee[1,8,9]*

1 Department of Population and Family Health, Columbia University Mailman School of Public Health, New York, New York, United States of America, 2 Ghana Health Service, Health Directorate, Nkwanta South, Ghana, 3 Department of Pediatrics and Child Health, University for Development Studies School of Medicine, Tamale, Ghana, 4 Department of Pediatrics and Child Health, Tamale Teaching Hospital, Tamale, Ghana, 5 Department of Anaesthesiology and Intensive Care, University for Development Studies School of Medicine, Tamale, Ghana, 6 Nkwanta South Municipal Health Directorate, Nkwanta South, Ghana, 7 Melbourne EpiCentre, Department of Medicine at Royal Melbourne Hospital, University of Melbourne and Melbourne Health, Melbourne, Australia, 8 Division of Infectious Diseases and Immunology, Department of Medicine, NYU Grossman School of Medicine, New York, New York, United States of America, 9 Health Unit, International Rescue Committee, New York, New York, United States of America

* megan.coffee@nyulangone.org

**Data Availability Statement:** The data underlying this study are available in Open Science Foundation (OSF) [76]: Crews C, Angwaawie P, Abdul-Mumin

## Abstract

Infection prevention and control (IPC) measures safeguard primary healthcare systems, especially as the infectious disease landscape evolves due to climate and environmental change, increased global mobility, and vaccine hesitancy and inequity, which can introduce unexpected pathogens. This study explores the importance of an "always-on," low-cost IPC approach, focusing on the role of natural ventilation in health facilities, particularly in low-resource settings. Ambient carbon dioxide ($CO_2$) levels are increasingly used as a measure of ventilation effectiveness allowing for spot checks and targeted ventilation improvements. Data were collected through purposive sampling in Northern Ghana over a three-month period. Levels of $CO_2$ ppm (parts per million) were measured by a handheld device in various healthcare settings, including Community-Based Health Planning and Services (CHPS) facilities, municipal and teaching hospitals, and community settings to assess ventilation effectiveness. Analyses compared $CO_2$ readings in community and hospital settings as well as in those settings with and without natural ventilation. A total of 40 facilities were evaluated in this study; 90% were healthcare facilities and 75% had natural ventilation (with an open window, door or wall). Facilities that relied on natural ventilation were mostly community health centers (60% vs 0%) and more commonly had patients present (83% vs 40%) compared with facilities without natural ventilation. Facilities with natural ventilation had significantly lower $CO_2$ concentrations ($CO_2$ ppm: 663 vs 1378, p = 0.0043) and were more likely to meet international thresholds of $CO_2$ < 800 ppm (87% vs 10%, p = <0.0001) and $CO_2$ < 1000 ppm (97% vs 20%, p = <0.0001). The adjusted odds ratio of low $CO_2$ in the natural facilities compared with non-natural were: odds ratios, OR (95% CI): 21.7 (1.89, 247) for $CO_2$ < 800 ppm, and 16.8 (1.55, 183) for $CO_2$ < 1000 ppm. Natural ventilation in these facilities was consistently significantly associated with higher likelihood of low $CO_2$

A, Yabasin I b, Attivor E, Coffee M. Ambient carbon dioxide concentrations in health facilities in Ghana [Internet]. OSF; 7 May 2024. Available: osf.io/ 92thw.

**Funding:** The authors received no specific funding for this work.

**Competing interests:** The authors have declared that no competing interests exist.

concentrations. Improved ventilation represents one cost-effective layer of IPC. This study highlights the continuing role natural ventilation can play in health facility design in community health care clinics. Most health facilities met standard $CO_2$ thresholds, particularly in community health facilities. Further research is needed to optimize the use of natural ventilation. The use of a handheld devices to track a simple metric, $CO_2$ levels, could improve appreciation of ventilation among healthcare workers and public health professionals and allow for them to target improvements. This study highlights potential lessons in the built environment of community primary health facilities as a blueprint for low-cost, integrated multi-layer IPC measures to mitigate respiratory illness and anticipate future outbreaks.

## Introduction

Infection prevention and control (IPC) measures safeguard primary healthcare, the backbone of healthcare systems. An always-on, low-cost IPC approach is essential when diagnoses may be delayed and threats unrecognized, which may occur more frequently as emerging diseases caused by climate change—along with environmental changes, increased mobility, and vaccine hesitancy and inequity—transforms the infectious disease landscape, particularly in low resource settings.

Effective IPC relies on layers of protection. The "hierarchy of control" or Swiss Cheese model layers engineering controls (such as building design and ventilation), administrative controls (triage, isolation, quarantine, IPC training and sick days), and personal protective equipment (PPE) to improve IPC effectiveness and reduce dependence on single factors, such as unpredictable supply chains and staffing changes [1–4]. Ventilation serves as one such always-on layer of protection against pathogens, particularly respiratory but also others, including aerosolized enteric pathogens. Improved ventilation has been used in health facilities against mycobacterium tuberculosis, a persistent healthcare setting risk and the single greatest cause of mortality from infectious diseases [5]; common respiratory viruses, including SARS-CoV-2 and influenza [3, 6]; vaccine-preventable infections, such as measles and varicella [7]; and emerging climate-change-related infections, including zoonoses such as avian influenza or even fungal infections, including in laboratory settings [8, 9]. Improved ventilation outside of healthcare also reduces influenza cases and school absences due to illness [10, 11]. Outbreak response may involve costly outlay for constructing isolation units parallel to permanent primary health infrastructure, but effectiveness relies on accurate diagnosis (underdiagnosis may expose healthcare providers; overdiagnosis may expose patients isolated together) [12–14]. Enhancing existing, always-on, cost-effective protective measures provides another layer of healthcare protection.

As recommended by the WHO [15], improved ventilation has long been achieved through natural ventilation and incorporated into health facility design. Natural ventilation is the flow of outdoor air into and out of indoor spaces, relying on wind and buoyancy due to temperature differences [16]. Such ventilation has long been known to reduce airborne tuberculosis (TB) transmission leading to purposeful design in health facilities to maximize this advantage [5, 17–19]. Natural ventilation can occur in several ways. Engineering solutions can manipulate natural ventilation where heating, ventilation, and air conditioning (HVAC) are not feasible [20]. Louvered windows allow fresh air to protect against rain and debris [21]. This airflow can rely on a) single-sided ventilation (one-sided open windows), b) cross-ventilation (horizontal airflow), c) stack ventilation (vertical airflow relying on differences in temperature and

wind to allow air to enter from below and exit from above), and d) top-down ventilation [22]. Displacement ventilation (or stack ventilation) allows air to enter from below and be extracted from above, offering a potential measure, where negative-pressure isolation rooms are needed [23].

However, natural ventilation can depend on many factors including wind patterns, corridor design, ceiling height, floor level, and unplanned impediments like bookcases or boxes blocking airflow [24, 25]. Windows may be closed for privacy (in labor and delivery ward), temperature regulation (neonatal care), security, and to control mosquitoes, wildfire smoke, or industrial pollution [26–28]. Climate change, whose impact will be felt the most among those least responsible for its burden, will increase the need for an always-on IPC approach for emerging diseases in lower resourced settings, but may also affect the use of natural ventilation, given temperature changes and risk of wildfire and air pollution [29]. A clearer understanding of natural ventilation's application and limitations will enable its more effective use in evolving contexts.

Natural ventilation does not provide all the benefits of mechanical ventilation, as it is not as easily controlled and does not incorporate filtration. Natural ventilation in clinics may allow the entry of outdoor hazards, including disease-carrying mosquitoes and particulate matter, as well as smaller, infectious agents like fungal spores or even Brucella from an abattoir [30–34]. Larger hazards can be mitigated through the implementation of screens and fans. Reducing $CO_2$ also deters mosquitoes, by reducing their attraction to the indoors, but filtering smaller hazards remains challenging [31, 35, 36]. Moreover, in West Africa, Harmattan and its seasonal dusty winds are associated with an increase in infections, including pneumococcal infections, trachoma, or conjunctivitis, as dry air and dust particulates can irritate mucosal passages and are more difficult to filter and reduce the benefit of natural ventilation [37–43].

Ventilation has often been quantified through ACH (Air Changes per Hour) which measures the number of times the air within a space is replaced with fresh air in one hour. ACH can be calculated by the decay in carbon dioxide ($CO_2$) levels in a space over time. Increasingly steady-state $CO_2$ levels measured in occupied indoor spaces have been used as a standalone measure of ventilation, although the limitations of this approach should be acknowledged [44, 45]. Certain national entities and professional organizations have established target levels, with most aiming for 1000 parts per million (ppm) as the upper limit of normal and others targeting 800 ppm [46–50]. Such measurement reflects occupancy and captures some of the risks associated with exposure to higher numbers of individuals (and potentially infections) in an indoor space, as more people generate more $CO_2$. Such $CO_2$ levels will also depend on the size of the person and activity, which may be correlated with age and sex, but which may not correspond to infection risk [45, 51–53]. Further, the duration of viability of any specific infectious particle in the air is not captured by the $CO_2$ level. It should be noted that as climate change intensifies, ambient environmental $CO_2$ levels will increase and the lowest achievable health facility levels will rise slightly; in 2022, the average global atmospheric $CO_2$ level was 417 ppm [54]. Although imperfect, such ventilation metrics can identify areas with a greater risk in rural and primary care facilities where closer monitoring and filtration systems are not feasible. This metric creates a tool that can be communicated with clinicians and other healthcare decision makers to identify weak points in healthcare systems and add extra layers of targeted IPC protection.

Ghana's robust primary healthcare system relies on decentralized community-based health planning and service (CHPS) facilities to bring patients' first contact with a health provider closer to their communities. Referrals are made for higher-level care at municipal, regional, and teaching hospitals if needed [55–57]. Strengthening primary health care involves preventing and treating illness at the community level, thereby decreasing resource constraints at

higher levels of care and reducing the spread of nosocomial infectious diseases in densely populated health facilities [58]. Improving the built environment of healthcare facilities at each level of care is a low-cost intervention for pursuing this goal and identifying weak points could reduce risk. In the Ghanaian context, previous research has assessed the levels of $CO_2$ in hospital facilities [59]; however, it is crucial to conduct further research at the CHPS level to understand the initial point of contact of an infectious disease with a larger health system and the community outside formal health facilities. Outdoor waiting areas can be better utilized and mechanized approaches can be considered where natural ventilation alone was insufficient [60, 61]. Lowering $CO_2$ levels could reduce concerns about the spread of airborne or respiratory infections; this would allow for a greater emphasis on addressing other service delivery needs [51, 62–64]. Here, we evaluated the $CO_2$ levels and the design of primary, secondary, and tertiary health facilities, considering the first contact areas between those needing triage and treatment in Ghana's health system, to determine whether built environments which utilized natural ventilation were correlated with adequate and lower $CO_2$ levels.

## Materials and methods

Data points were captured using purposive sampling during interactions with health facilities in the Gushegu and Nkwanta South municipalities as well as the Tamale Metropolitan Assembly, over a 3-month period from June 16th to September 15th, 2022, see catchment area in Fig 1. To improve resource efficiency, a convenience sampling frame limited site selection to the three municipalities that were accessible to researchers within the 3-month research period. Nested purposive sampling within the chosen municipalities selected a heterogenous distribution of primary, secondary, and tertiary healthcare facilities that were 1) accessible to researchers within a 1-hour travel radius from the metropolitan center of the municipality and 2) actively seeing patients on the day of the visit, so that data taken would more accurately approximate the average user experience during daytime operations.

In Nkwanta South, the average temperature is 27°C and ranges from 22°C to 34°C. The average humidity was 80%, which ranged from 56% to 90% during the time period. Windspeed average measured at Tamale airport was 4mph, with a maximum of 12mph during this time period. The rainy season in Ghana is from April to mid-October [65]. Sampling included a variety of patient densities and types of facility areas, including CHPS facilities, municipal hospitals, and teaching hospitals. Field notes recorded qualitative assessments to accompany each data point, noting the facility (healthcare/non-healthcare), type of location (e.g., conference center, obstetric theater), a rough estimate of the number of people within the enclosed space, a short description of the built environment, highlighting whether windows or doors were open and whether the structure had open walls, and if there was any artificial ventilation, such as a fan or air-conditioning unit (AC); however, the direction of airflow (AC recirculation, outside air, or fan direction) was not recorded. Some patient waiting areas had half wall structures and were referred to as having open walls; the structures feature walls that extend only partway up, often to waist or chest height, and are covered by a roof above.

A $CO_2$ monitor manufactured by Aranet4 Technology was used to record $CO_2$ (in ppm) at each data collection point. Data collectors were briefed on standard $CO_2$ monitoring procedure, which involved waiting for 20 minutes for the device to calibrate to the room environment before obtaining a reading. Monitors were placed 50 cm away from people to avoid contamination by direct exhaled breath, at head height, away from windows, doors, and air supply openings, and at approximately the center of the room. Readings were taken when the room inhabitants included data collectors, health staff, and clients; therefore, the total number of inhabitants in each room may not be representative of how many would be present during

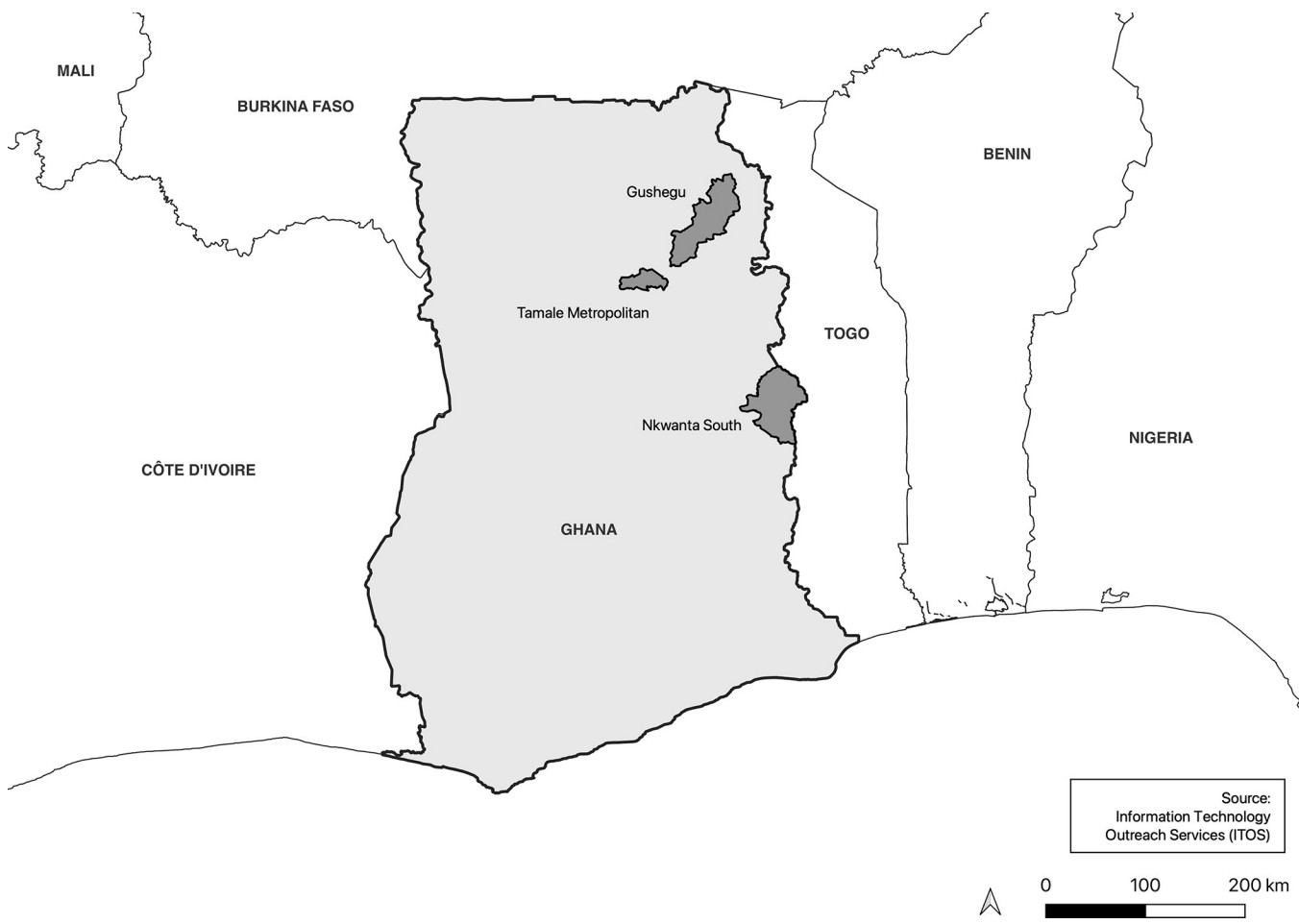

**Fig 1. Catchment areas for data collection in three municipalities in Ghana.** Map created in QGIS using the "Ghana—Subnational Administrative Boundaries" base layer supplied by OCHA West and Central Africa and Ghana Statistical Services [66] and under the CC BY 3.0 IGO license.

standard practice, contributing to potentially higher concentrations of people than normal. $CO_2$ levels were recorded per indoor space, but also calculated per person and per person in excess of atmospheric levels (the per person share of the amount above the atmospheric 417 ppm in 2022).

## Statistical analysis

Variables were summarized as count (%) and mean (SD), separately between the open and closed windows. The summary measures of the variables were compared between the two groups using two-sample t-test or Wilcoxon rank sum test for continuous variables and Chi-square test or Fisher exact test for categorical variables. $CO_2$ concentration values below 800 ppm or 1000 ppm were used as a cut-off to identify facilities with low $CO_2$ level [46–50]. Multivariate binomial logistic regression with $CO_2$ levels (Low vs High) as primary outcome and Window/Door/Wall (WDW) status (Open vs Close) as main exposure, was used to assess the independent association between the WDW and $CO_2$ levels while adjusting for potential covariates. Variable redundancy was done using divisive principal component variable clustering to reduce multicollinearity between variables (SAS Certified Statistical Business Analyst Using SAS 9: Regression and Modeling, SAS Institute Inc., Cary, NC, USA). The final model

results are expressed as odds ratio, OR (95% Confidence Interval (CI). WDW was considered significantly associated with low $CO_2$ levels if the 95% CI did not contain 1. All analyses were done using SAS 9.4 (SAS Institute, Inc., Cary, NC) and conclusions made at 5% significance level.

### Data storage and confidentiality

Data were entered into an Excel spreadsheet and made accessible to all study personnel. Location information was de-identified to the municipal level, including only field notes on the built environment and type of setting. Permission to collect $CO_2$ readings in health facilities was granted by the administration at the Tamale Teaching Hospital and the Municipal Health Service Directorates in the Gushegu and Nkwanta South municipalities. The research was determined at New York University Langone not to involve human subjects and was therefore exempt from the human subjects' regulations as there was no identification of, data collected from, communication with, interpersonal contact with, or interactions with any individual patients.

### Inclusivity in global health research

Additional information regarding the ethical, cultural, and scientific considerations specific to inclusivity in global research is included in the S1 Checklist.

## Results

### Overall facility characteristics

The basic-level structure for CHPS facilities in Ghana is a two-room structure with an attached pavilion, which, in most cases, serves as a waiting area for patients and an area where child welfare clinic services are conducted. All buildings in the CHPS facilities can be naturally ventilated using windows with adjustable louvres and doors that serve as cross-ventilation openings. The facility windows were covered with nets to prevent the entry of mosquitoes, flies, and other potential disease vectors. Curtains on windows and doors provide privacy. In facilities with consistent power supply, ceiling fans are switched on in most rooms when the weather is hot. No air conditioners were installed at CHPS facilities or health centers that were included in the study. Owing to the compact design of CHPS facilities, furniture, such as cupboards, fridges, notice boards, and shelves, are sometimes located close to windows, creating congestion that tends to impede the free ventilation flow into rooms. No devices that increase $CO_2$ readings, such as those for cooking, heating, or sterilization of medical equipment were observed. Facilities with improved design, with multiple rooms and better spacing, located in CHPS zones that were not surveyed.

Healthcare data were disaggregated by healthcare facilities into readings from municipal and teaching hospitals and CHPS facilities, as shown in Table 1. Across multiple rooms within their physical structures, CHPS facility and hospital readings averaged 679.8 ppm (n = 18) and 973.0 ppm (n = 18), respectively. CHPS facilities were more likely to have open windows or doors in patient rooms and open-air waiting room spaces than hospitals, where field notes recorded the frequent use of wall-mounted air conditioning (AC) or fans. Of the ten facility spaces observed to have no natural ventilation, seven had AC. AC can be set to recirculation or fresh air intake, but device settings could not be observed. With very few exceptions, the lowest quartile of $CO_2$ readings was recorded in either open-air structures or rooms with open windows. All open-window and/or open-door facilities had $CO_2$ values lower than 1000 ppm except one value of 1040 ppm in a hospital laboratory. All the CHPS facilities had $CO_2$ levels at

**Table 1. Descriptive statistics on the count of location type, average ppm in each location typology, and average number of people in each location typology.**

| Location of reading | Count of Location | Average ppm | Average # of people in each location |
|---|---|---|---|
| **CHPS facility** | **18** | **679.8** | **7.1** |
| | | **(range: 530–855)** | |
| Consulting room | 7 | 662.6 | 4.4 |
| Delivery room | 2 | 611.0 | 8.5 |
| Reception room | 1 | 572.0 | 8.0 |
| Stockroom | 1 | 621.0 | 4.0 |
| Waiting area | 7 | 740.6 | 9.7 |
| **Hospital** | **18** | **973.0 (453–2793)** | **12.8** |
| Canteen | 1 | 542.0 | 5.0 |
| Waiting area | 2 | 521.0 | 40.5 |
| Consulting room | 2 | 1177.5 | 7.0 |
| ICU | 3 | 715.7 | 13.3 |
| Recovery room | 2 | 826.5 | 13.5 |
| Laboratory | 1 | 1040.0 | 13.0 |
| Hospital administrator's office | 1 | 1445.0 | 4.0 |
| Conference room | 2 | 1713.5 | 12.0 |
| In-patient ward | 1 | 559.0 | 4.0 |
| Operating theatre | 2 | 1164.0 | 5.5 |
| Reception room | 1 | 976.0 | 7.0 |
| **Non-healthcare setting** | **4** | **982.0 (602–1530)** | **37.3** |
| Emergency dispatch center | 1 | 1053.0 | 3.0 |
| Church | 1 | 602.0 | 76.0 |
| Conference center | 1 | 1530.0 | 42.0 |
| Gym | 1 | 743.0 | 28.0 |
| **Grand Total** | **40** | **842.0** | **12.7** |

or below 855ppm, with 16 of 18 surveyed having values 800 ppm or less. Hospital inpatient wards, including intensive care units (ICUs), had values less than 1000 ppm. In contrast, the highest $CO_2$ reading quartile was recorded in rooms with no open windows or doors, with fans and ACs. Operating theatres, where windows may be closed for sterility, had values above 1000ppm. Conference or administrative offices that relied on AC also had higher $CO_2$ values. A hospital conference room had the highest reading overall at 2793 ppm with 15 occupants, closed windows and doors, air conditioning and fan turned on. In contrast, a hospital obstetric recovery room had the lowest reading at 453 ppm, despite 10 occupants; it had eight windows open and three fans running.

Data collectors recorded readings in patient and non-patient areas of the healthcare facilities. The areas where patients were seen, such as consulting rooms or waiting areas, had lower average $CO_2$ levels (748.3 ppm) than areas where only staff congregated, such as the clinic laboratory, conference room, and hospital canteen, which had higher average $CO_2$ levels (1150.1 ppm). Furthermore, the concentration (ppm) per person can serve as an indicator of ventilation effectiveness. CHPS facilities averaged fewer people per room (7.1) compared to that in hospitals (12.8) and had a lower average ppm/person (114.9) than that of hospitals (125.5).

Data collectors also recorded some (n = 4) readings in non-healthcare facilities, specifically in a gym, hotel conference center, administrative emergency ambulance dispatch center, and church, and found widely varying results. Using open windows and doors, the church, holding a large number of people (76), was better equipped to maintain lower $CO_2$ levels at 602 ppm.

**Table 2. Comparison of indoor spaces by clinical CO$_2$ ppm threshold (800ppm)*.**

| Variables | | Overall | CO$_2$ ppm ≥ 800 | CO$_2$ ppm <800 | P value |
|---|---|---|---|---|---|
| | N (%) | 40 | 13 (33) | 27 (67) | |
| | # of people | 12.7 (14.7) | 13.3 (10.9) | 12.4 (16.4) | 0.19 |
| Window/Door/Wall | | | | | |
| | *Closed* | 10 (25%) | 9 (69%) | 1 (4%) | <0.0001 |
| | *Open* | 30 (75) | 4 (31) | 26 (96) | |
| Patients present | | | | | 0.16 |
| | *Healthcare workers only* | 7 (18%) | 4 (31%) | 3 (11%) | |
| | *Patients present* | 29 (73) | 7 (54) | 22 (81) | |
| | *Not a healthcare facility* | 4 (10) | 2 (15) | 2 (7) | |
| Facility level | | | | | 0.0328 |
| | *Community health center* | 18 (45%) | 2 (15%) | 16 (59%) | |
| | *Hospital* | 18 (45) | 9 (69) | 9 (33) | |
| | *Non-healthcare* | 4 (10) | 2 (15) | 2 (7) | |

*Data values in mean (SD) or count (%). Percentages may not total 100% due to rounding.

In contrast, the conference center, with no open windows but with mechanical ventilation (air-conditioning), had much higher CO$_2$ levels, with fewer people (42) at 1530 ppm.

## Facility characteristics by CO$_2$ levels

Tables 2 and 3 show the distribution of the facilities variables by level of CO$_2$ levels. A total of 40 facility readings were included in this survey. Of these, 90% were in healthcare facilities and 75% relied on natural ventilation. Overall, the average number of persons present in each space measured was 13 people. The facilities relying on natural ventilation (defined by open windows, doors, or walls) achieved considerably better rates of acceptable CO$_2$ levels, below the standard thresholds of 800 ppm (2a) or 1000 ppm (2b). Community health facilities, which

**Table 3. Comparison of indoor spaces by clinical CO$_2$ ppm threshold (1000ppm)*.**

| Variables | | Overall | CO$_2$ ppm ≥ 1000 | CO$_2$ ppm <1000 | P value |
|---|---|---|---|---|---|
| | N (%) | 40 | 9 (23) | 31 (77) | |
| | # of people | 12.7 (14.7) | 12.8 (12.1) | 12.6 (15.5) | 0.68 |
| Window/Door/Wall | | | | | |
| | *Closed* | 10 (25%) | 8 (89%) | 2 (6%) | <0.0001 |
| | *Open* | 30 (75) | 1 (11) | 29 (94) | |
| Patients present | | | | | 0.08 |
| | *Healthcare workers only* | 7 (18%) | 3 (33%) | 4 (13%) | |
| | *Patients present* | 29 (73) | 4 (44) | 25 (81) | |
| | *Not a healthcare facility* | 4 (10) | 2 (22) | 2 (6) | |
| Facility level | | | | | 0.005 |
| | *Community health center* | 18 (45%) | 0 (0%) | 18 (58%) | |
| | *Hospital* | 18 (45) | 7 (78) | 11 (36) | |
| | *Non-healthcare* | 4 (10) | 2 (22) | 2 (7) | |

*Data values in mean (SD) or count (%). Percentages may not total 100% due to rounding.

**Table 4. Comparison of $CO_2$ per person and the $CO_2$ per person beyond the atmospheric level compared between those with and without natural outdoor air ventilation (windows, doors, and/or walls open).**

| Variables | | Overall | No Natural Ventilation | Natural Ventilation | P value |
|---|---|---|---|---|---|
| | N (%) | 40 | 10 (25) | 30 (75) | |
| | # of people | 12.7 (14.7) | 11.8 (11.6) | 13.0 (15.7) | 0.95 |
| | $CO_2$ PPM per person | 118.7 (84.9) | 187.0 (120.6) | 96.0 (55.0) | 0.0324 |
| | Excess $CO_2$ PPM per person | 56.6 (60.3) | 125.3 (85.4) | 33.7 (21.2) | 0.0017 |
| | $CO_2$ PPM overall level | 842.0 (439.9) | 1378 (597.4) | 663.3 (132.2) | 0.0043 |
| Patient present | | | | | 0.0189 |
| | *Healthcare workers only* | 7 (18) | 4 (40) | 3 (10) | |
| | *Patients present* | 29 (73) | 4 (40) | 25 (83) | |
| | *Not healthcare facility* | 4 (10) | 2 (20) | 2 (7) | |
| Facility level | | | | | 0.0019 |
| | *Community health center* | 18 (45) | 0 (0) | 18 (60) | |
| | *Hospital* | 18 (45) | 8 (80) | 10 (33) | |
| | *Non-healthcare* | 4 (10) | 2 (20) | 2 (7) | |

*Data values in mean (SD) or count (%).

all relied on natural ventilation, had the highest rate of acceptable $CO_2$ levels. Most community health facilities (88.9%) met the 800ppm threshold, while all community health facilities met the 1000ppm threshold. In hospitals, only half of readings met the 800ppm threshold and 61% met the 1000ppm threshold. Most locations where patients were present met the 800ppm (76%) and 1000pm (86%) thresholds. Health facility locations where patients were not present (such as administrative offices and conference rooms) often did not meet the threshold standards; only 43% met 800pm and 57% met 1000pm, though only 7 such sites were studied.

## Association between natural outdoor air ventilation and low $CO_2$ levels

Table 4 shows the distribution of the variables between facilities with natural and non-natural ventilation. Compared to non-natural ventilation facilities, those with natural ventilation have significantly low $CO_2$ concentrations ($CO_2$ ppm: 663 vs 1378, p = 0.0043; $CO_2$ ppm per person: 96 vs 187, p = 0.0043; Excess $CO_2$ ppm per person: 34 vs 125, p = 0.0017; $CO_2 < 800$ ppm: 87% vs 10%, p = <0.0001; $CO_2 < 1000$ ppm: 97% vs 20%, p = <0.0001). In addition, facilities with natural ventilation when compared with facilities without natural ventilation were more commonly community health centers (60% vs 0%), had patients present (83% vs 40%), and on average had slightly higher number of people (13 vs 12). After adjusting for facility level patients present, and number of people, the adjusted odds ratio of low $CO_2$ in the natural facilities compared with non-natural were: odds ratios, OR (95% CI): 21.7 (1.89, 247) for $CO_2 < 800$ ppm, and 16.8 (1.55, 183) for $CO_2 < 1000$ ppm (Table 5). Natural ventilation is consistently significantly associated with higher likelihood of low $CO_2$ concentrations.

## Discussion

Although ventilation has been recognized as critical to health infrastructure for over a century and architecture has long capitalize on its benefit, clinicians have few tools to recognize and assess ventilation in their health facilities and have few studies act to document and guide in low resource settings. Natural ventilation is often quite effective, as seen here, but is an invisible layer protecting against infections, unlike PPE. Having a simple metric to pinpoint areas at

**Table 5. Logistic regression for the association between natural ventilation and low CO2 ppm.**

| Variables | | Adjusted OR (95% CI) | |
|---|---|---|---|
| | | $CO_2 < 800$ ppm | $CO_2 < 1000$ ppm |
| **Natural ventilation** | | | |
| | *Yes* | 26.6 (2.24, 315) | 18.12 (1.77, 186) |
| | *No (reference)* | | |
| **Facility level** | | | |
| | *Community health center* | 0.69 (0.01, 36.6) | 8.6 (0.07, 100) |
| | *Hospital* | 0.47 (0.02, 11.7) | 1.53 (0.05, 43.4) |
| | *Non-healthcare (reference)* | | |
| **Number of people** | | 0.98 (0.92, 1.05) | 0.99 (0.93, 1.08) |

higher risk for poor ventilation and disease transmission can identify needed modifications. These can prompt simple steps such as moving bookshelves blocking a window or ensuring cross ventilation with two open doors. It can mean focusing investments on areas without natural ventilation alternatives, such as by prioritizing mechanical ventilation in operating theatres to avoid open windows contaminating surgical fields and where masking already reduces respiratory risk.

Here the low $CO_2$ levels in CHPS facilities (average of 679.8 ppm) and higher $CO_2$ levels in hospitals (average of 973 ppm) demonstrates that there are lessons to be learned in the built environments of these smaller community facilities. There was a statistically significant difference in means of $CO_2$ levels between built environments with natural ventilation and without, wherein those with natural ventilation saw lower levels of $CO_2$ on average. This effect was not simply because these facilities had fewer individuals present. The overall levels in the CHPS facilities were all below indoor ventilation standards for high quality air in Japan (1000 ppm) and well below the US workplace upper limit (5000 ppm). It is promising to see that resource-constrained facilities, such as rural Ghana, can consistently achieve these results and create an important layer of protection [44, 67].

Investment in air-conditioning alone for indoor climate control may otherwise result in using recycled air and less ventilation, leading to a deceptively cool office space, while overlooking the benefits in reducing respiratory disease transmission. The data here, showing higher levels of $CO_2$ in air-conditioned spaces, is a reminder that AC, especially when on recirculation and not with fresh air intake, is not a substitute for natural ventilation, where consciously designed open-air waiting spaces and air flow can achieve needed ventilation. Natural ventilation will not protect against all risks, but safeguards such as louvered windows and screens (and lowered $CO_2$ itself reduces mosquito attraction) can reduce hazards [35, 61, 68].

The $CO_2$ metric allows healthcare workers to pinpoint high-risk areas for undetected respiratory pathogen transmission which might have been unrecognized. This risk is not just in patient care areas, but often among staff- only areas in poorly ventilated break rooms, laboratories, conference rooms, canteens, and offices, where there may be less attention paid to the danger of nosocomial spread among the staff. Laboratory ventilation is an essential layer of protection, well known in tuberculosis (TB) laboratories, but the commitment needs to be maintained as microbes are inherently undiagnosed and new pathogens may arrive unexpectedly as climate change and mobility may expand the range of pathogens, from Burkholderia pseudomallei to fungal infections. Moreover, past outbreaks in hospitals have been traced back to healthcare worker chains of transmission both in SARS-CoV-2 outbreaks but also in other types of disease transmission, such as with Ebola [69, 70]. These outbreaks often occur in breakrooms where no patients are seen. The data here highlight the importance of designing

well-ventilated spaces across the entire facility, including in healthcare worker only areas; the least well-ventilated spaces in the sample included conference room, administrative offices, and clinician consultation rooms, highlighting the need for attention to healthcare worker-only areas.

To better address ventilation, more data can pinpoint areas at risk and highlight already well-designed facilities. This study presents an initial picture of qualitative assessments of built environments in health facilities and quantitative measures of $CO_2$ using standardized measurements in these facilities. However, further research is warranted to elucidate best practices for clinicians and public health practitioners to better maximize the role of ventilation in the built environment. Further studies can more closely study respiratory—as well as aerosolized enteric—disease spread to better recognize the impact of natural and artificial ventilation in health facilities. These initial results speak to the quality of the design of existing facilities and are promising in terms of documenting $CO_2$ ppm levels and the circulatory capacity of healthcare spaces.

The use of ventilation in health facilities progressed from miasma to germ theory during the 19th century, often motivated by the persistent threat of TB and later by influenza in the 20th century [18, 71]. In the 21st century, epidemics of respiratory pathogens, such as SARS-CoV-1 and SARS-CoV-2, underscored the need for equitable ventilation solutions as technology advanced toward higher-cost HVAC. Here $CO_2$ monitoring is a reminder of the importance of natural ventilation and designed spaces in reducing the risk of nosocomial transmission, which is important to understand as climate change may impact the application of these measures as it affects temperatures, weather patterns, and disease risks.

## Limitations

The metric used, $CO_2$ ppm, does not fully capture the risk of respiratory pathogen transmission. More research is needed to understand how to use $CO_2$ readings effectively in clinical environments to improve IPC and respond to different pathogen risks.

Data collection was limited by the availability of facilities within the sampling frame; data points included $CO_2$ readings, occupancy figures, and notes on built environments. The statistical analyses only highlight that $CO_2$ levels were found to be low with natural ventilation. This is reassuring but does not mean that artificial ventilation could not achieve the same results. Cost-effectiveness necessitated the use of purposive sampling in the study, thereby limiting the representativeness of the findings, but future studies should consider probability sampling on a larger scale to conduct more extensive causal analyses on the associations between $CO_2$ levels and built environments. Improving ventilation requires an understanding of how room height and cross ventilation affects draft patterns, how AC settings can affect ventilation, how outdoor climate influences ambient indoor $CO_2$ levels, how room function and occupant activities alter expelled $CO_2$, and how the building design itself encourages or discourages airflow through intentionally placed windows, hallways, and doors. Future analyses can consider seasonality with temperature and wind direction changes, but here readings were limited to June to September, owing to data collector availability. Data from colder or Harmattan months may be less reassuring in community health facilities if windows are closed. Data can also follow variation of $CO_2$ with different occupancy levels throughout the day or with modifications in the indoor environment. Further data on the structure size, layout, and other air quality measures will add to the picture. Future research can build on this exploration into an under-researched body of literature by examining the possibility of making several $CO_2$ monitors available for year-long data collection and environmental modification.

## Conclusion

Resource-constrained countries have struggled to implement global IPC standards and maintain PPE supply chains in outbreaks and emergencies [6, 14, 72–75]. A broad understanding of healthy environments should prioritize effective and robust additive layers that are "always-on", especially as climate change and mobility may create unexpected infectious risks. These approaches do not necessarily require "high tech" or costly solutions, but instead can utilize what is tried and true. The known importance of natural ventilation can be highlighted with hand-held $CO_2$ readers used by clinicians and public health professionals. These metrics can increase the appreciation of existing solutions and guide the use of simple adjustments to improve ventilation and reduce nosocomial respiratory disease transmission. Public health and clinical professionals should broaden their appreciation of the infrastructure strategies implemented globally. The use of $CO_2$ ppm metrics can shine light on ventilation as an IPC strategy and inspire practical adjustments to improve IPC in health facilities.

The data underlying this study are available in Open Science Framework (OSF) [76].

## Supporting information

**S1 Checklist. Inclusivity in global research.**
(DOCX)

## Acknowledgments

We appreciate help and guidance from Dr Rachel Moresky as well as from all our colleagues at the Ghana Health Service, University for Development Studies School of Medicine, Tamale Teaching Hospital, NYU Langone, including the Section for Global Health and NYU AMPATH (Academic Model Providing Access to Healthcare) Ghana, Columbia University Mailman School of Public Health, the International Rescue Committee, and AIM-AHEAD (Artificial Intelligence/Machine Learning Consortium to Advance Health Equity and Researcher Diversity).

## Author Contributions

**Conceptualization:** Megan P. Coffee.

**Data curation:** Cecilia Crews, Paul Angwaawie, Alhassan Abdul-Mumin, Megan P. Coffee.

**Formal analysis:** John Dibato, Megan P. Coffee.

**Investigation:** Cecilia Crews, Paul Angwaawie, Alhassan Abdul-Mumin, Iddrisu Baba Yabasin, Evans Attivor.

**Methodology:** Cecilia Crews, Paul Angwaawie, Megan P. Coffee.

**Project administration:** Cecilia Crews, Paul Angwaawie, Alhassan Abdul-Mumin, Iddrisu Baba Yabasin.

**Resources:** Cecilia Crews, Paul Angwaawie, Alhassan Abdul-Mumin, Iddrisu Baba Yabasin, Evans Attivor, Megan P. Coffee.

**Supervision:** Cecilia Crews, Alhassan Abdul-Mumin, Megan P. Coffee.

**Validation:** Cecilia Crews, Paul Angwaawie, Alhassan Abdul-Mumin, Iddrisu Baba Yabasin, Evans Attivor.

**Visualization:** Cecilia Crews.

**Writing – original draft:** Cecilia Crews, Megan P. Coffee.

**Writing – review & editing:** Cecilia Crews, Paul Angwaawie, Alhassan Abdul-Mumin, Iddrisu Baba Yabasin, Evans Attivor, John Dibato, Megan P. Coffee.

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
