## [Decision Letter · Decision Letter 0]

27 Mar 2024

PGPH-D-24-00416

Assessing ventilation through ambient carbon dioxide concentrations across multiple healthcare levels in Ghana

Dear Megan,

Thank you for submitting your manuscript to PLOS Global Public Health. After careful consideration, we feel that it has merit but does not fully meet PLOS Global Public Health’s publication criteria as it currently stands. Therefore, we invite you to submit a revised version of the manuscript that addresses the points raised during the review process.

We look forward to receiving your revised manuscript.

Kind regards,

Collins Otieno Asweto, PhD

Academic Editor

Journal Requirements:

2. Please include a complete copy of PLOS’ questionnaire on inclusivity in global research in your revised manuscript. Our policy for research in this area aims to improve transparency in the reporting of research performed outside of researchers’ own country or community. The policy applies to researchers who have travelled to a different country to conduct research, research with Indigenous populations or their lands, and research on cultural artefacts. The questionnaire can also be requested at the journal’s discretion for any other submissions, even if these conditions are not met.  Please find more information on the policy and a link to download a blank copy of the questionnaire here: https://journals.plos.org/globalpublichealth/s/best-practices-in-research-reporting. Please upload a completed version of your questionnaire as Supporting Information when you resubmit your manuscript.

3. Please provide separate figure files in .tif or .eps format only and remove any figures embedded in your manuscript file. Please also ensure all files are under our size limit of 10MB.

4. We ask that a manuscript source file is provided at Revision. Please upload your manuscript file as a .doc, .docx, .rtf or .tex.

5. Please amend your detailed Financial Disclosure statement. This is published with the article. It must therefore be completed in full sentences and contain the exact wording you wish to be published.

If you did not receive any funding for this study, please simply state: “The authors received no specific funding for this work.

6. Some material included in your submission may be copyrighted. According to PLOS’s copyright policy, authors who use figures or other material (e.g., graphics, clipart, maps) from another author or copyright holder must demonstrate or obtain permission to publish this material under the Creative Commons Attribution 4.0 International (CC BY 4.0) License used by PLOS journals. Please closely review the details of PLOS’s copyright requirements here: PLOS Licenses and Copyright. If you need to request permissions from a copyright holder, you may use PLOS's Copyright Content Permission form.

Potential Copyright Issues:

Fig 1: please (a) provide a direct link to the base layer of the map (i.e., the country or region border shape) and ensure this is also included in the figure legend; and (b) provide a link to the terms of use / license information for the base layer image or shapefile. We cannot publish proprietary or copyrighted maps (e.g. Google Maps, Mapquest) and the terms of use for your map base layer must be compatible with our CC-BY 4.0 license. 

Additional Editor Comments (if provided):

Reviewers' comments:

Reviewer's Responses to Questions

**Comments to the Author**

1. Does this manuscript meet PLOS Global Public Health’s publication criteria? Is the manuscript technically sound, and do the data support the conclusions? The manuscript must describe methodologically and ethically rigorous research with conclusions that are appropriately drawn based on the data presented.

Reviewer #1: Yes

Reviewer #2: Yes

2. Has the statistical analysis been performed appropriately and rigorously?

Reviewer #1: No

Reviewer #2: Yes

3. Have the authors made all data underlying the findings in their manuscript fully available (please refer to the Data Availability Statement at the start of the manuscript PDF file)?

Reviewer #1: Yes

Reviewer #2: Yes

4. Is the manuscript presented in an intelligible fashion and written in standard English?

Reviewer #1: Yes

Reviewer #2: Yes

5. Review Comments to the Author

Reviewer #1: There are a few issues that need to be clarified in the paper.

1. It is stated that purposive sampling was used. It would be better if an explanation is provided to indicate why this method of sampling was used. Overall, the justifications used for the type of sampling used. What prompted the selection of the specific study locations in the three municipalities as identified in the study?

2. The parameters for the logistic regressions and discussions of the logistics results are missing. It is mentioned that this was carried out, but this has not been indicated anywhere in the paper. In line 213-214 there is a discussion that may allude to these results but from the tables shared it is not possible to locate table one.

3. The use of descriptive statistics as the only statistical analysis may not be adequate to validate the conclusions arrived at based on the data and the overall research question of the paper. Consider adjusting the statistical approach used.

Reviewer #2: I consider the evaluation of ventilation systems in healthcare facilities and hospitals to be of utmost importance, and I commend the authors for their diligent research and assessment. The methodology employed in this study is robust, and the findings offer valuable insights and recommendations.

6. PLOS authors have the option to publish the peer review history of their article (what does this mean?). If published, this will include your full peer review and any attached files.

**Do you want your identity to be public for this peer review?** For information about this choice, including consent withdrawal, please see our Privacy Policy.

Reviewer #1: No

Reviewer #2: No

---

## [Decision Letter · Decision Letter 1]

1 Jul 2024

Assessing ventilation through ambient carbon dioxide concentrations across multiple healthcare levels in Ghana

PGPH-D-24-00416R1

Dear Coffee,

We are pleased to inform you that your manuscript 'Assessing ventilation through ambient carbon dioxide concentrations across multiple healthcare levels in Ghana' has been provisionally accepted for publication in PLOS Global Public Health.

Best regards,

Collins Otieno Asweto, PhD

Academic Editor

Reviewer's Responses to Questions

**Comments to the Author**

1. If the authors have adequately addressed your comments raised in a previous round of review and you feel that this manuscript is now acceptable for publication, you may indicate that here to bypass the “Comments to the Author” section, enter your conflict of interest statement in the “Confidential to Editor” section, and submit your "Accept" recommendation.

Reviewer #1: All comments have been addressed

Reviewer #2: All comments have been addressed

2. Does this manuscript meet PLOS Global Public Health’s publication criteria? Is the manuscript technically sound, and do the data support the conclusions? The manuscript must describe methodologically and ethically rigorous research with conclusions that are appropriately drawn based on the data presented.

Reviewer #1: Yes

Reviewer #2: Yes

3. Has the statistical analysis been performed appropriately and rigorously?

Reviewer #1: Yes

Reviewer #2: Yes

4. Have the authors made all data underlying the findings in their manuscript fully available (please refer to the Data Availability Statement at the start of the manuscript PDF file)?

Reviewer #1: Yes

Reviewer #2: Yes

5. Is the manuscript presented in an intelligible fashion and written in standard English?

Reviewer #1: Yes

Reviewer #2: Yes

6. Review Comments to the Author

Reviewer #1: All my comments and issues have been addressed.

Reviewer #2: Great job with all the additions, it creates more clarity

7. PLOS authors have the option to publish the peer review history of their article (what does this mean?). If published, this will include your full peer review and any attached files.

**Do you want your identity to be public for this peer review?** For information about this choice, including consent withdrawal, please see our Privacy Policy.

Reviewer #1: No

Reviewer #2: No
